# Optimized Small Waterbird Detection Method Using Surveillance Videos Based on YOLOv7

**DOI:** 10.3390/ani13121929

**Published:** 2023-06-09

**Authors:** Jialin Lei, Shuhui Gao, Muhammad Awais Rasool, Rong Fan, Yifei Jia, Guangchun Lei

**Affiliations:** 1School of Ecology and Nature Conservation, Beijing Forestry University, Beijing 100083, China; leijialinbjfu@foxmail.com (J.L.); fanrong1119@bjfu.edu.cn (R.F.); 2Birdsdata Technology (Beijing) Co., Ltd., Beijing 100083, China; gaoshuhui_bupt@163.com; 3Burewala Sub Campus, University of Agriculture Faisalabad, Vihari 61010, Pakistan; moosa33381@uaf.edu.pk

**Keywords:** conservation, frame, SimAM attention module, small object detection, surveillance video, waterbird monitoring, YOLO v7 algorithm

## Abstract

**Simple Summary:**

Waterbird monitoring is crucial for conservation and management strategies in wetland ecosystems. There is limited research on using deep learning techniques for small waterbird detection from real-time surveillance videos. This study describes an improved detection method by adding an extra prediction head, SimAM attention module, and sequential frame to YOLOv7, termed as YOLOv7-waterbird for the real-time video surveillance devices to identify attention regions and monitor waterbirds.

**Abstract:**

Waterbird monitoring is the foundation of conservation and management strategies in almost all types of wetland ecosystems. China’s improved wetland protection infrastructure, which includes remote devices for the collection of larger quantities of acoustic and visual data on wildlife species, increased the need for data filtration and analysis techniques. Object detection based on deep learning has emerged as a basic solution for big data analysis that has been tested in several application fields. However, these deep learning techniques have not yet been tested for small waterbird detection from real-time surveillance videos, which can address the challenge of waterbird monitoring in real time. We propose an improved detection method by adding an extra prediction head, SimAM attention module, and sequential frame to YOLOv7, termed as YOLOv7-waterbird, for real-time video surveillance devices to identify attention regions and perform waterbird monitoring tasks. With the Waterbird Dataset, the mean average precision (mAP) value of YOLOv7-waterbird was 67.3%, which was approximately 5% higher than that of the baseline model. Furthermore, the improved method achieved a recall of 87.9% (precision = 85%) and 79.1% for small waterbirds (defined as pixels less than 40 × 40), suggesting a better performance for small object detection than the original method. This algorithm could be used by the administration of protected areas or other groups to monitor waterbirds with higher accuracy using existing surveillance cameras and can aid in wildlife conservation to some extent.

## 1. Introduction

Since the 1960s, the international community has continuously recognized the core value of wetland ecosystems in the sustainable development of biodiversity, the human economy, and society, and has recognized that migratory waterbirds that rely on wetlands are vital resources of the entire ecosystem [1]. Wetland management, especially waterbird conservation, requires governments to promulgate laws and policies to coordinate actions with other countries. As the country implementing the Ramsar Convention, China has fulfilled its international commitments. One of the achievements is the development of long-term national field ecological observation stations that ensure the integration of monitoring and supervision through technology using acoustic monitors and surveillance videos [2], promoting ecology and conservation in the era of big data.

Knowing the accurate location of wildlife enables optimized management and conservation [2]. Compared to the traditional methods of waterbird monitoring (Direct Count Method with binocular or telescope), real-time video surveillance devices have many advantages in terms of monitoring duration, storage, and verifiability [3], thus providing more accurate abundance and distribution information. The development of camera-based observation systems and, advancements in deep learning applications, offer new tools to study animal conservation research [4,5]. However, the surge in remote surveillance studies has contributed to a growing recognition of the challenges associated with screening vast amounts of image data for target species. Thus, automated recognition methods rather than human eyes are vital [6].

Deep learning (DL) or convolutional neural networks (CNN) have recently been increasingly used to investigate the possibilities of automated recognition in the conservation field [7,8]. This method, unlike other high-resolution image capture devices for surveillance requires near-real-time detection, which means higher computing speed and accuracy. One-stage object detection methods, such as You Only Looking Once (YOLO) [9], single-shot multibox detector (SSD) [10], and Retinanet [11], have a better speed performance than the two-stage methods. In addition to inheriting the advantages of the original YOLO model, YOLO v7 [12] has better detection accuracy and faster inference speed because of its more sophisticated network structure and training strategy [13].

Among recently published and recognized works [14], object detection based on DL remains a fundamental task in the field of artificial intelligence. Few studies focusing on small object detection in remote sensing images have been designed for wildlife and livestock surveillance [15], despite some studies applied to detect tiny objects, such as people, boats, or traffic accidents [16]. Small waterbird detection using the latest version of YOLO (v7) still seems to be particularly challenging, especially when faced with an object less than 40 × 40 pixels [17]. Surveillance video images are frequently unpredictable, due to the possibilities of changing weather, fixed position, illumination directions, object distances, and image compression. Thus, the waterbird detection rate may undergo a sharp decline [18]. In this study, we propose an enhanced algorithm, termed YOLOv7-waterbird, for small waterbird detection in real-time surveillance videos. This algorithm incorporates an additional prediction head, SimAM attention module, and sequential frame to YOLOv7. To build the Waterbird Dataset, we collected and checked over 8500 images containing multi-taxon species from a total of 3000 videos of over 10 s in length that were extracted with the aid of cameras fixed at six wetlands. We evaluated the performance of YOLOv7-waterbird by testing it on the Waterbird Dataset and comparing it to the baseline model. Our results indicate that YOLOv7-waterbird achieved a higher mean average precision (mAP) value of 67.3% and better performance for small object detection compared to the original method. The proposed algorithm has the potential to enable the administration of protected areas or other groups to monitor waterbirds with higher accuracy using existing surveillance cameras and could aid in wildlife conservation efforts.

## 2. Materials and Methods

### 2.1. Data Collection and Filtering

Following avian ecology investigations that used digital video recording (DVRs) devices, remote DVRs were fixed for surveillance in six wetlands that are waterbird habitats, namely Chenhu (Wuhan, Hubei province, China), Tiaozini 720 High tidal Roost site (Yancheng, Jiangsu province, China), Zhanjiang Mangrove Forest wetlands (Zhanjiang, Guangdong province, China), Nandagang wetlands (Cangzhou, Hebei province, China), West Dongting Lake (Hanshou, Hunan province, China), and Guangyang Island (Chongqing municipality, China). The open network video interface forum (ONVIF) surveillance camera (iDS-2DF8C84015XS-AFW/SP(T2B), Haikang, Zhejiang, China) had no less than 8 million effective pixels, supported autonomous cruise (cruising tasks can be customized), and had 7.5–300 mm focal length and 40 times optical zoom. The ONVIF-DVRs were also equipped with a built-in GPS and Beidou module to support precise positioning of the field of view, lens pointing, and automatic time adjustment and were accessible via IP addresses [17]. Each camera was fixed at 8 m or 30 m above the ground according to field observation requirements and programmed to scan the area at a fixed time interval. The camera collected data during the period from 30 October 2022 to 15 February 2023, covering the wintering and migration seasons of waterbirds in monitoring wetlands. The key image frames of a fixed sequence (3 s) were extracted from the videos.

A total of 3000 videos of over 10-s length were extracted, and over 8500 images containing multi-taxon species (over 70 species, Appendix A) were collected and manually screened to ensure data quality. Eighty percent of the high-quality images (1920 × 1080) were chosen for use in the training process, and the remainder were assigned for testing.

### 2.2. Data Annotation

The lableImg tool was used to label the collected pictures, ensuring:

A: COCO annotation format [18].

B: Removal of images that were difficult to recognize by the human eye (smaller than 5 × 5 pixels); annotations under five numbers were ignored. Those crowded over five were tagged as a single frame.

C: During the labeling process, open-source YOLOv7 was used to pre-label the images, and the pre-labeled label results were manually checked and modified where necessary.

Example of annotated images are as follows (Figure 1):

### 2.3. YOLOv7-Waterbird Algorithm

#### 2.3.1. Detection Framework

The YOLO network’s excellent characteristics yielded an improved YOLOv7-waterbird, which is shown in Figure 2. This consisted of three parts. First, the backbone of YOLOv7 was adopted to extract the feature maps. The SimAM attention module was integrated into the neck and head sections (an extra head for tiny object detection), to enable the network to recognize the difference between the target and background, reduce false recall, and improve detection performance [19]. Because the dataset contained many very small instances, we added a small target prediction head to the head to build a sequential frame feature fusion that better combined the information of shallow features and had a higher output resolution, which greatly improved the detection effect of the model for small targets. Finally, we use the tracking results to optimize the detection results for the front and rear frames, making the detection results more stable (Figure 2).

#### 2.3.2. Extra Prediction Head

Generally, the images in the Waterbird Dataset had the characteristics of rather small targets and a large meadow or water surface interference, which created great challenges for the model. As shown in Figure 3, a target frame size below 40 × 40 accounted for 42.94% of the cases. The original YOLOv7 contained three detection output heads. The largest feature map size is 1/8 of the original input image, taking 640 input images as an example; the largest feature map size is 80 × 80, mapped back to the input image, so each mapped pixel covers eight pixels of the original image, which is not very convenient for detecting small objects. We added a new small object detection head branch, combined with shallower features and greater resolution, and could thus output the feature map resolution, which is 1/4 of the original image. The extra head can bring low-level, high-resolution feature information into the feature fusion layer, making the added prediction head more sensitive to small objects, thereby improving the prediction effect of the model for small targets.

#### 2.3.3. SimAM Attention Module

Attention modules are widely used in DL to enhance feature extraction and to focus on useful target objects. Current mainstream attention modules are mainly channel-based 1D attention modules, airspace-based 2D attention modules, and a combination of these two. The principle is that by the addition of weight modules for different channels and locations, different treatment for different channels and different locations may be achieved. Weight modules often require the implementation of additional subnetworks; therefore, additional parameters must be added. Existing research has primarily focused on the effects of attention by combining two attention mechanisms. The two types of attention in the human brain often work together; therefore, SimAM proposes an attention module with a unified weight. SimAM calculates the importance of neurons by measuring their linear separability without introducing additional parameters [19]. The attention mechanism involves finding important neurons by measuring the linear separability between neurons and assigning a higher priority to these neurons. Our experiments in this study show how the introduction of the SimAM nonparametric attention mechanism in the YOLOv7 structure helps the model extract the feature information of the objects more effectively during the detection process without increasing the original network parameters.

In particular, to differentiate the importance of neurons and successfully achieve attention, an energy function is defined to determine the linear separability between the target neuron and all other neurons in the same channel. The final energy function is provided by
$$e_{t\;}\left(\omega_{t},b_{t},y,x_{i}\right)=\frac{1}{M-1}\sum_{i=1}^{M-1}{\left(-1-\left(\omega_{t}x_{i}+b_{t}\right)\right)}^{2}+{\left(1-\left(\omega_{t}t+b_{t}\right)\right)}^{2}+{\lambda \omega }_{t}^{2}$$

The transformation weights and bias are expressed as follows:$$\omega_{t}=\frac{2\left(t-\mu_{t}\right)}{{\left(t-\mu_{t}\right)}^{2}+2\sigma_{t}^{2}+2\lambda }$$
$$b_{t}=-\frac{1}{2}\left(t+\mu_{t}\right)\omega_{t}$$
where $\mu_{t}=\frac{1}{M-1}\sum_{i=1}^{M-1}x_{i}$ and $\sigma_{t}^{2}=\frac{1}{M-1}\sum_{i=1}^{M-1}{\left(x_{i}-\mu_{t}\right)}^{2}$ are mean and variance calculated over all neurons except t in that channel, respectively. Given the assumption that all pixels in a single channel follow the same distribution, mean and variance can be calculated and reused for all neurons on that channel, by avoiding iteratively calculating μ and σ for each position by significantly reducing the computation costs. The minimum energy equation is as follows:$$e_{t}^{*}=\frac{4\left(\sigma^{2}+\lambda \right)}{{\left(t-\hat{\mu }\right)}^{2}+2\sigma^{2}+2\lambda }$$
where $\hat{\mu }=\frac{1}{M}\sum_{i=1}^{M}x_{i}$ and $\sigma^{2}=\frac{1}{M}\sum_{i=1}^{M}{\left(x_{i}-\hat{\mu }\right)}^{2}$.

#### 2.3.4. Sequential Frame Add

The video-based target detection task differs from the single-frame image target detection task, which requires the detection of the target in each frame of the video sequence. The detection accuracy of a single frame decreases sharply, owing to the poor quality of the target in some frames such as blurred image and occlusion, considering that attributes such as shape and scale change with the movement of the target. Because videos contain more temporal information than static images, most video object detection algorithms use temporal information to enhance detection performance [20]. The different integration methods can be roughly divided into the following four categories [21] (Figure 4):

A: Combination of single-frame target detection and tracking algorithm

B: Use of optical flow information to perform feature propagation in the time dimension

C: Enhancing Keyframe Features Using Attention Mechanism

D: Others (LSTM, 3D convolution)

The combination of single-frame target detection and target tracking is a relatively common and simple video target detection method. After carefully reviewing the dataset, we found that waterfowl exhibited diverse movements and postures (Figure 5), which sometimes changed quickly. Therefore, associating the detection information of multiple consecutive frames through target tracking can improve the detection accuracy for a single-frame picture.

Following Henriques et al., (2014) [22] for the video target detection algorithm TCNN, we pushed back the video 3 s prior to the current frame for target detection and chose the frame with the highest score as the initial tracking frame for target tracking, using high-speed tracking with Kernelized Correlation Filters (KCF). We then tracked the current frame and combined it with the tracking frame and the current frame using Non-Maximum Suppression (NMS) for fusion (Figure 6).

### 2.4. Model Training

YOLOv7 was pretrained on the COCO dataset [16], which is a large-scale object detection and captioning dataset. Because the proposed YOLOv7-waterbird and YOLOv7 share a backbone and most of the head, we divided our dataset into a training dataset (6599 images) and a test dataset (1643 images). The training dataset contained the open-source data of CoCo and ImageNet and was randomly spliced, as the birds accounted for a large proportion of pixels. We trained the model only on the waterbird training dataset for 200 epochs, and the first three epochs were used for the warm-up. The model was trained and inferred on eight NVIDIA 2080Ti GPUs, and Stochastic Gradient Descent (SGD) was utilized as the optimizer with a default weight decay of 0.0005 and a momentum of 0.937. The size of the input image for our model was very large, with a side length of 1280.

To enhance the dataset diversity and prevent overfitting, data augmentation was considered to extend the dataset, in which random combinations of transformation operations involving translation, scaling, rotation, and dithering were utilized. Translation, scaling, and rotation can increase the number of labeled objects in images [7]. Meanwhile, maritime scenarios were enriched by color dithering.

### 2.5. Performance Evaluation

The detection performance of the learned models was evaluated using the interpolated average precision (AP), which is the commonest performance index for evaluating detection accuracy. We applied the AP metric using an Intersection over Union (IoU) threshold of 0.5 and 0.95, and the mean average precision (mAP), which is the average of all the thresholds of the IoU in the range [0.50:0.95], as the criterion for judging the target detection. IoU is defined mathematically by Equation (1), where $B_{p}\cap B_{\mathrm{gt}}$ and $B_{p}\cup B_{\mathrm{gt}}$ denote the intersection and union of the predicted ($B_{p}$) and ground truth ($B_{\mathrm{gt}}$) bounding boxes, respectively.
(1)$$\mathrm{IoU}=\frac{\mathrm{Area}\left(B_{p}\cap B_{\mathrm{gt}}\right)}{\mathrm{Area}\left(B_{p}\cup B_{\mathrm{gt}}\right)}$$

AP was calculated as the area under the precision-recall graph with a predetermined detection threshold, as described in Equations (2)–(4) (where TP is the true positive, that is, the number of birds correctly detected; FP is the false positive, that is, the number of incorrect detections; and FN is the false negative, that is, the number of ground truth birds undetected).
(2)$$\mathrm{Precision}\;=\frac{\mathrm{TP}}{\mathrm{TP}+\mathrm{FP}}+\mathrm{FP}\;$$
(3)$$\mathrm{Recall}\;=\frac{\mathrm{TP}}{\mathrm{TP}+\mathrm{FN}}$$
(4)$$\mathrm{AP}=\int_{0}^{1}P\text{©}\mathrm{dR}$$

## 3. Results

### 3.1. Model Performance

We compared the four methods on the Waterbird Dataset for mAP and AP, which were evaluated by setting the IoU threshold to 0.5 (listed in Table 1). The results for the different methods showed an increasing trend in AP as more modules were added. The highest mAP value was 67.3% when all the three modules were added, whereas the lowest mAP value was 63.9% when only YOLOv7 was used. As more modules were added, the AP (IoU:0.5) value gradually increased from 86.4% to 91.7%. In particular, the first stage presented a relatively marked increase of 4.1%, demonstrating the effectiveness of adding an additional detection head.

Judging from the actual effect after adding the extra prediction head, the ability of the model to detect small targets remarkably improved (Figure 7a). After adding the SimAM attention module, the ability of the model to distinguish between the foreground and background was greatly improved (Figure 7b). After adding the timing information, the retrieval of some birds with strange poses in the current frame also improved (Figure 7c).

### 3.2. Precision and Recall

For small object detection, we certainly require precision and usually consider the recall rate when the precision is greater than 80%. As shown in Table 2, the recall rate gradually increased (YOLOv7-addhead-atten-time > YOLOv7-addhead-atten > YOLOv7-addhead > YOLOv7), indicating that YOLOv7-addhead-atten-time can detect more small waterbird targets than the other methods, whereas YOLOv7 alone presented the worst performance, as expected. We also separately calculated the recall of small waterbirds and other birds (defined as pixels larger than 40 × 40) to check model performance for small waterbirds (Table 2). Although the recall of small waterbird detection was far below that of the others (~20%), the percentage increases were slightly higher at 5.1% and 2.8% from the baseline to the improved model, respectively (Figure 8).

## 4. Discussion

### 4.1. Impact Factors

In this study, an improved object detection framework was developed to automatically recognize waterbirds in surveillance videos. The improved model achieved an mAP of 67.3%, which is approximately 5% higher than that of the baseline model (YOLOv7). The application of deep learning (DL), particularly the YOLO framework, for identifying waterbirds in real-time surveillance videos has been limited in previous studies [15]. Waterbird data has traditionally been collected using techniques such as aerial vehicles [23], drones [24], and camera traps [25], resulting in datasets with varying quality and deep learning algorithm applications. Consequently, direct comparisons with other DL frameworks are difficult to make, thereby weakening comparability. Therefore, the discussion categorically focuses on the potential impact factors within the data quality and different modules. Specifically, the error results were as follows: (i) the assumed target was blocked by other objects, including aquatic plants, gullies on the everglade, or just another waterfowl, and (ii) background complexity. Because of their distinct dietary niches, waterbirds are divided into different foraging groups, such as sedge/grass foragers, tuber feeders, fish, clam, and invertebrate eaters, thus demonstrating diversity in habitat selection and use [23]. That is, the potential habitats could be marshes, swabs, mudflats, shallow waters, and many other types rather than just water, which in our case was proven by the images collected from the Waterbird Dataset. In conclusion, objects would cause erroneous detection (false positives) more often if they appear on a complex background or if their body color camouflages the background. (iii) In some cases, it was difficult to detect and recognize birds, even with the human eye. This situation might occur in species that rarely move and use stalked foraging strategies (e.g., pond herons, Figure 9).

Therefore, a sophisticated camera with better performance is the best alternative [7].

### 4.2. Limitations and Prospects

Because of the development level of current convolutional neural network, existing models cannot meet the counting ability in all scenarios, particularly for dense small targets, and the performance of target detection algorithms has always been unsatisfactory [16]. Although our algorithm has made some progress in small object detection (recall improved from 74% to 79.1% when precision is 85%), there is still a huge gap compared to larger targets (recall from 91.0% to 93.8% when precision is 85%), which will continue to be one of the tasks we explore in the future. From a practical point of view, due to the different distances of the birds from the camera, the difference in body size sometimes cannot be reflected in the image results. For example, a Falcated Teal (*Mareca falcata*) that is closer may look larger in the picture than a Eurasian Spoonbill (*Platalea leucorodia*) that is farther away. Therefore, we pay more attention to the proportion of pixels in the final image than to the type. As mentioned in the results (Table 2), our team is more confident in the detection performance when the proportion of target pixels reaches 40 × 40 or more. Another point worth noting is the monitoring distance limit. The distance between the monitoring equipment and the target affects the pixel size of the target and, consequently, the outcome of the surveillance [24]. To ensure accurate and reliable results, it is recommended that the observation distance for large water birds, such as the Oriental Stork and Common Crane, be within 1 km range when the focal length is around 400 mm and obstruction is minimal. For geese and ducks, a range of 500 m is recommended, while for small birds such as sandpipers, the observation range should be within 300 m.

Secondly, while considering video transmission speed and saving bandwidth, common surveillance devices use different image compression algorithms [25], which have a certain impact on the details of the image during the video recording process, thus affecting the accuracy of the algorithm. For the specific optimization of image compression and transmission algorithms, it will also benefit the use of deep learning for intelligent automatic monitoring [26]. Besides, wetland monitoring hardware currently relies primarily on fixed equipment. While mobile monitoring equipment provides some degree of flexibility, using a combination of high towers and mobile monitoring equipment enables more comprehensive large-scale quantitative statistics and detailed observation of key areas. Edge computing with video monitoring can reduce the need for human intervention and enable faster response times by facilitating real-time data processing and analysis. It can also improve overall system performance by reducing network latency through the reduction of data transmission to the cloud. As a result, the combination of edge computing with video monitoring has the potential to improve the accuracy, efficiency, and scalability of wildlife monitoring efforts [27]. In summary, this method is viable with manual review for waterbird monitoring and data collection. However, an efficient hardware combined with edge monitoring is crucial to achieve the dream of a fully automated monitoring system in the near future.

## 5. Conclusions

In this paper, we tried to establish an improved detection method for the real-time video surveillance devices for waterbird monitoring, which have become a more common dataset in the wetland nature reserves. The small target size and variable posture data collected by these surveillance devices brings great challenges to general detection models. To deal with this situation, first, we built a new large-scale dataset of common waterbirds captured by real-time surveillance cameras in different wetlands. The dataset contains 60,582 labels in 8242 images of the 62 main kinds of waterbirds in the Wuhan, Yancheng, Hanshou, Zhanjiang, Chongqing, and Cangzhou city wetlands, making it the largest dataset for small waterbird detection in China. Second, an extra prediction head, SimAM attention module, as well as sequential frame were added to YOLOv7, and then, YOLOv7-waterbird algorithm was proposed and tested. As a result, the new model showed a better performance in both AP and precision (mAP from 63.9% to 67.3% by contrast), which proves to be an improved method for waterbird detection. Although the precision seems to be lower compared with similar missions, such as boat detection, we hope that this will offer more experience in wildlife monitoring for developers and researchers.

## Figures and Tables

**Figure 1 animals-13-01929-f001:**
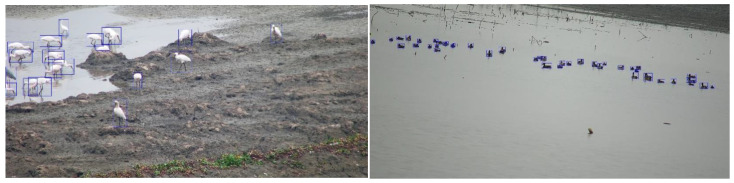
Samples of annotated images. Each bird in an image is marked as a blue frame.

**Figure 2 animals-13-01929-f002:**
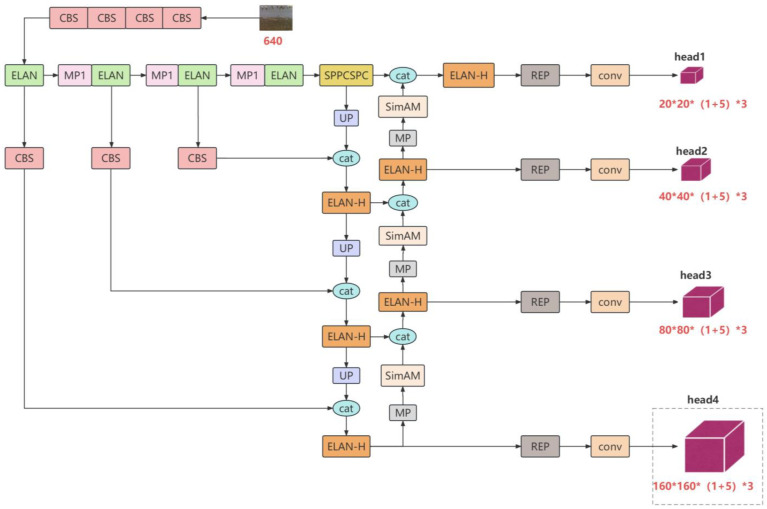
Architecture of YOLOv7-waterbird.

**Figure 3 animals-13-01929-f003:**
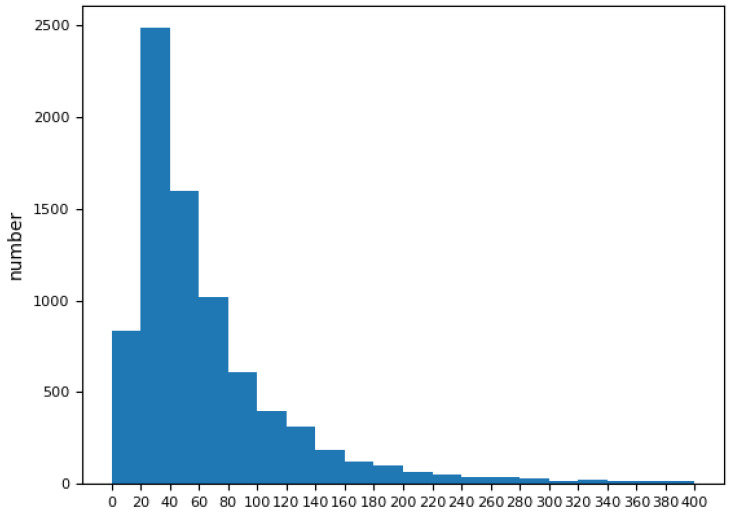
Number of labels of different pixel intervals.

**Figure 4 animals-13-01929-f004:**
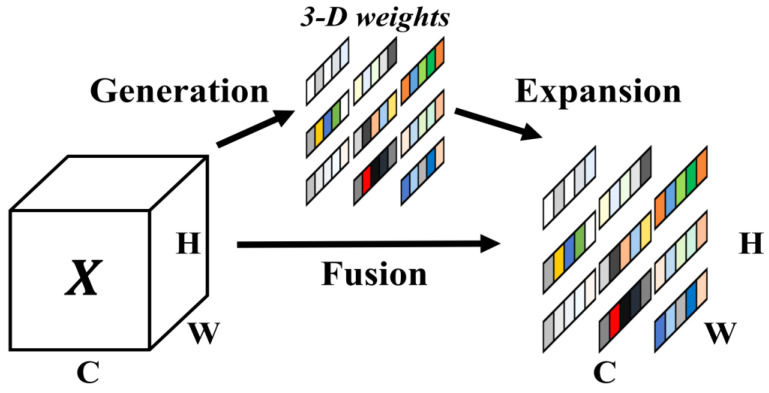
SimAM with full 3D weights for attention. Same color denotes that a single scalar is employed for each channel, for spatial location, or for each point on that feature [19].

**Figure 5 animals-13-01929-f005:**
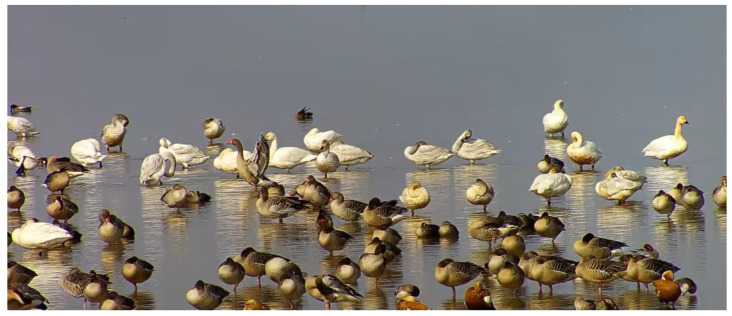
Waterbirds presenting different postures in the video, including rest, feeding, walking, or getting ready to fly.

**Figure 6 animals-13-01929-f006:**
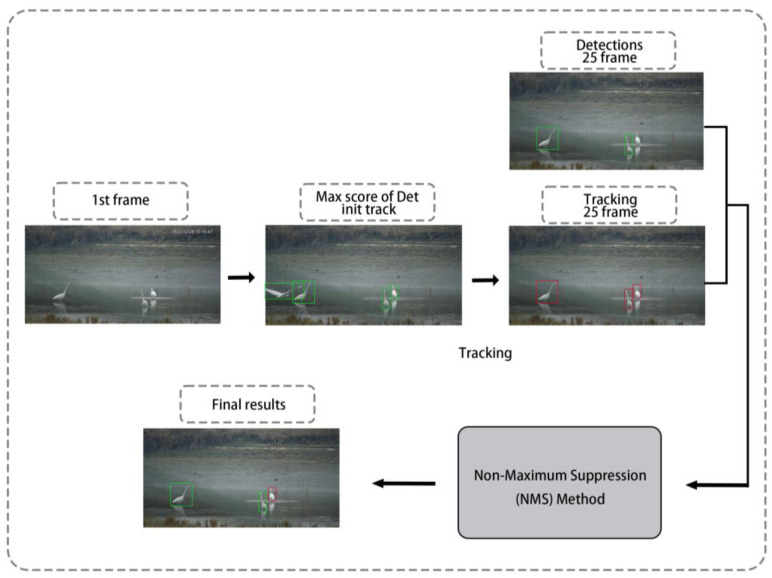
Sequential frame is added to the detection frame using NMS for fusion.

**Figure 7 animals-13-01929-f007:**
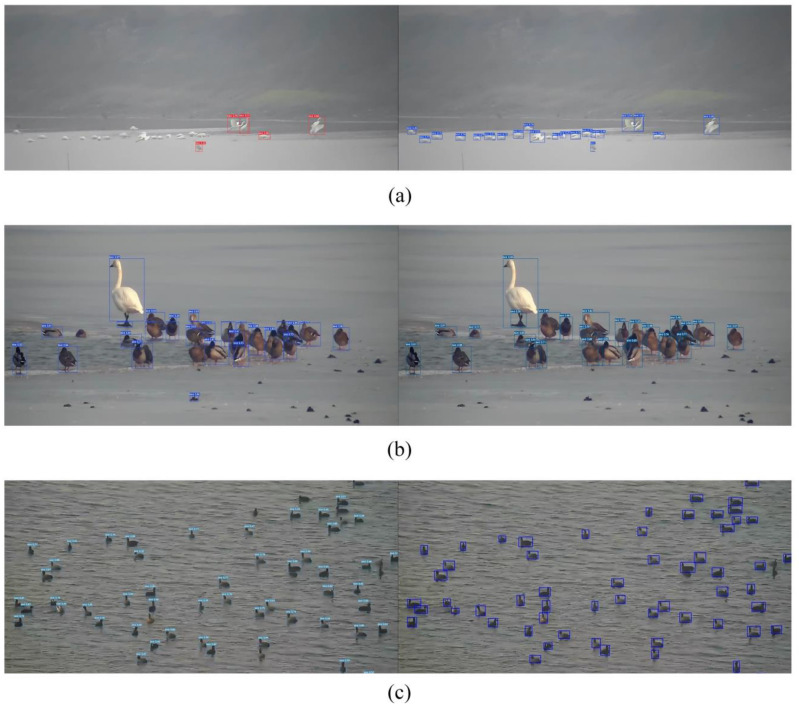
Detection results of YOLOv7-addhead method (**a**), YOLOv7-addhead-atten (**b**) and YOLOv7-addhead-atten-time (**c**).

**Figure 8 animals-13-01929-f008:**
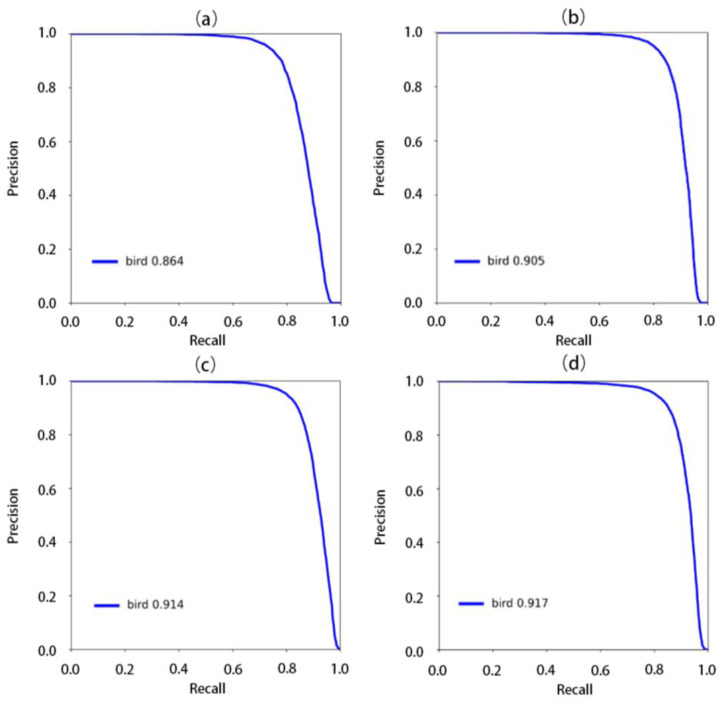
Precision-recall graphs of trained models for the intersection over union (IoU) threshold of 0.5: (**a**) YOLOv7; (**b**)YOLOv7-addhead; (**c**)YOLOv7-addhead-atten; and (**d**) YOLOv7-addhead-atten-time.

**Figure 9 animals-13-01929-f009:**
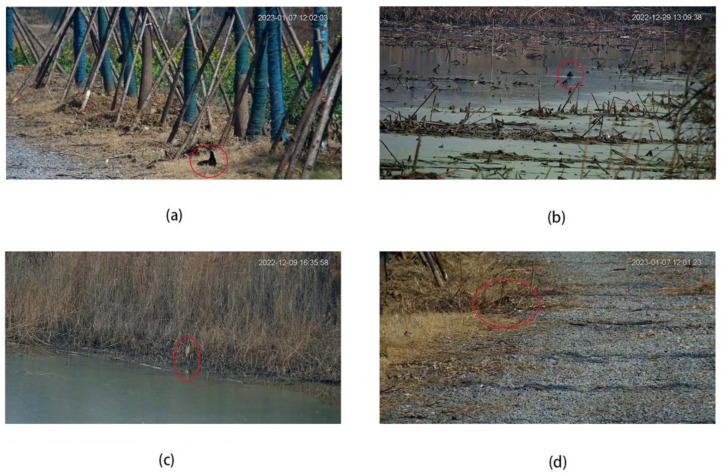
Erroneous detection results: (**a**) Two Crested Myna (*Acridotheres cristatellus*) overlapped; (**b**,**c**) no detection with complex background; (**d**) difficult to identify even with the human eye.

**Table 1 animals-13-01929-t001:** Test results of detection models on Waterbird Dataset.

Method	mAP (%)	AP (IoU:0.5) (%)
YOLOv7	63.9	86.4
YOLOv7-addhead	66.0	90.5
YOLOv7-addhead-atten	67.1	91.4
YOLOv7-addhead-atten-time	67.3	91.7

**Table 2 animals-13-01929-t002:** Test results of recall of all datasets, small waterbird (defined as pixel less than 40×40), and others when precision is 85.0%.

Method	All Data Recall (%)	Small Waterbird Recall (%)	Others Recall (%)
YOLOv7	79.8	74.0	91.0
YOLOv7-addhead	85.5	-	-
YOLOv7-addhead-atten	86.8	-	-
YOLOv7-addhead-atten-time	87.9	79.1	93.8

## Data Availability

Not applicable.

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
