# Peer review of "Optimized Small Waterbird Detection Method Using Surveillance Videos Based on YOLOv7"

_animals, 2023, doi:10.3390/ani13121929_

Round 1
Reviewer 1 Report
So far, using remote devices to collect acoustic and visual data of wildlife species is very popular in China, but data filtration and analysis techniques are still lack. In this manuscript, the authors proposed an improved detection method to identify attention regions and perform waterbird monitoring tasks. The authors showed that the average precision was around 5% higher comparing to the baselien model, and also the performance for small object detection was also improved. I think it's a very nice work which certainly meet the need especiall for the staff working in naturereserves. I have only several tiny questions as below:
1. In method part, cameras were fixed at different height, will it affect the results?
2. it says the research includes over 70 species, what are they, it will be nice to show the species name somewhere, maybe as a support information.
3. For species at different body size, does the precision varied? Is it possible to suggest, e.g. for species in which size we can have very high confidence.
Figure 2 is a bit difficult to read, you may need to increase the size of the text. Also for Figure 8
It's sufficient to read, but need some tiny modifications.
Author Response
Response to reviewer1:
We would like to express our sincere gratitude for your insightful review of our manuscript. We appreciate your time and effort in providing us with constructive feedback to improve our work. We have carefully considered all of your comments and suggestions, and we would like to respond to them point by point as follows:
Comment 1:
In method part, cameras were fixed at different height, will it affect the results?
Response:
Firstly, The installation height of the camera is limited by the infrastructure conditions of the installation site. In some areas, iron towers have been built before, so we can easily install the camera on it, while in some areas there is no infrastructure foundation, so we can only consider installing it on poles as the cost to install a tower is quite high.
Secondly, the cameras fixed at different height will mainly affect the observation range. Basically the cameras at 30 meters high can cover a range of 0-1000meters while the cameras at 8 meters high have a smaller range. Regardless of the installation height, those cameras can effectively capture images of waterfowl within its range, the identification results affected main by the distance from waterbird to the fixed location and boday size, rather than height. And all the images in the dataset have been manually screened to ensure the data quality (line 108), therefor we think the fix height will not affect the results.
Comment 2:
It says the research includes over 70 species, what are they, it will be nice to show the species name somewhere, maybe as a support information.
Response:
Thanks for the suggestion. We agree that including the names of the 70+ species studied would be beneficial for readers who are interested in the specifics of our research. , we have added a appendix table on line 485, and we also mentioned it at line 107.
Comment 3
For species at different body size, does the precision varied? Is it possible to suggest, e.g. for species in which size we can have very high confidence.
Response:
Thanks for the question. We would like to explain the question in two aspects:
In the case of the same distance from the camera, larger species have a larger image in the picture (such as Oriental white stork), so the detection effect is better, while shorebirds have a smaller image, and the effect will be slightly worse.
But from a practical point of view, as different distances of birds from the camera may affect the perceived size of the birds in the images, making it difficult to rely on body size alone for detection performance. For example, a Falcated Teal that is closer may look larger in the picture than a Spoonbill that is farther away. Therefore, we focus more on the proportion of pixels in the final image as an indicator of detection performance. As mentioned in our article's results (Table 2), our team is more confident in the detection performance when the proportion of target pixels reaches 40*40 or more.
Comment 4:
Figure 2 is a bit difficult to read, you may need to increase the size of the text. Also for Figure 8
Response:
Thank you for the feedback. We apologize for the inconvenience caused by the small size of the text in Figures 2 and 8. We have increased the size of the text in these figures for better readability in the final version of the manuscript.
Comment 5:
Comments on the Quality of English Language: It's sufficient to read, but need some tiny modifications.
Response:
Thanks for the suggestion. We have carefully reviewed the manuscript and made the necessary modifications to improve the language quality.
We appreciate your valuable comments, which have helped us to improve the manuscript's quality. Thank you again for your time and effort in reviewing our work.
Reviewer 2 Report
The authors present an interesting manuscript on mall waterbird detection identification. The bright side of the manuscript is that to provide some useful practical details on related topic. In this context, the study contributes to different fields. However, there are some missing points in the manuscripts. Therefore, I would like to make some suggestions to improve the quality of the paper as below:
Line 48: Please add such as sentence here “The development of camera-based observation systems and as advances in deep learning applications offer new tools to study animal conservation research (please add these references: 10.3390/ani10071207 and 10.1002/eap.2694)” I think, such sentence would make the bridge between the problem and objectives of the study stronger.
Lines 69-71: I think, this part of the manuscript should be re-phased. This part of the paper is important since authors should explain that what is the purpose of the study and what authors did. Please explain the aim of the study with 2-3 sentences here instead of single sentence.
Line 75: The time and the season of data collection period should be explained briefly. I mean, when the author collected data?
Line: 296: The Discussion section should be enriched with a more theoretical interpretation and relating the present results with additional concepts. For example, discussion section can be improved with the similar studies that focussed on bird/water bird detection. Moreover, the study results can be discussed in the framework of animal recognition and classification in broader context.
Author Response
Response to reviewer2:
We would like to express our sincere gratitude for your insightful review of our manuscript. We appreciate your time and effort in providing us with constructive feedback to improve our work. We have carefully considered all of your comments and suggestions, and we would like to respond to them point by point as follows:
Comment 1:
Line 48: Please add such as sentence here “The development of camera-based observation systems and as advances in deep learning applications offer new tools to study animal conservation research (please add these references: 10.3390/ani10071207 and 10.1002/eap.2694)” I think, such sentence would make the bridge between the problem and objectives of the study stronger.
Response:
Thanks for the suggestion, we agree with your suggestion to add a sentence that highlights the importance of camera-based observation systems and advances in deep learning applications for animal conservation research. We have carefully revised the manuscript and included the sentence and references in line 46-48.
Comment 2:
Lines 69-71: I think, this part of the manuscript should be re-phased. This part of the paper is important since authors should explain that what is the purpose of the study and what authors did. Please explain the aim of the study with 2-3 sentences here instead of single sentence.
Response:
Thanks for the suggestion. we agree that this part of the paper is crucial in explaining the aim of the study. In the revised manuscript, we have rephrased this section to provide a clearer and more concise explanation of the study's purpose:
“In this study, we propose an enhanced algorithm, termed YOLOv7-waterbird, for small waterbird detection in real-time surveillance videos. This algorithm incorporates an additional prediction head, SimAM attention module, and sequential frame to YOLOv7. To build the Waterbird Dataset, we collected and checked over 8,500 images containing multi-taxon species from a total of 3,000 videos of over 10 seconds in length that were extracted with the aid of cameras fixed at six wetlands. We evaluated the performance of YOLOv7-waterbird by testing it on the Waterbird Dataset and comparing it to the baseline model. Our results indicate that YOLOv7-waterbird achieved a higher mean average precision (mAP) value of 67.3% and better performance for small object detection compared to the original method. The proposed algorithm has the potential to enable the administration of protected areas or other groups to monitor waterbirds with higher accuracy using existing surveillance cameras and could aid in wildlife conservation efforts.”
Comment 3:
Line 75: The time and the season of data collection period should be explained briefly. I mean, when the author collected data?
Response:
Thank you for your feedback. We apologize for not providing a detailed description of the time and season of the data collection period. We collected data between October 2022 and February 2023, covering the wintering and migration seasons of waterbirds in monitoring wetlands. We have added this information to the revised manuscript in the line 102-104.
Comment 4:
Line: 296: The Discussion section should be enriched with a more theoretical interpretation and relating the present results with additional concepts. For example, discussion section can be improved with the similar studies that focussed on bird/water bird detection. Moreover, the study results can be discussed in the framework of animal recognition and classification in broader context.
Response:
Thank you for your helpful comments. We agree that a more in-depth theoretical interpretation would enhance the quality of the manuscript. We have taken the reviewer's comments into consideration and have enriched the discussion section with additional theoretical interpretations and concepts related to animal recognition and classification.
We highlighted the gap in detecting smaller targets compared to larger ones and discussed the need to explore this further in future research. We also pointed out the impact of different image compression algorithms on the accuracy of the algorithm and the need for specific optimization of image compression and transmission algorithms for intelligent automatic monitoring.
Lastly, we discussed the potential of edge computing with video monitoring in improving the accuracy, efficiency, and scalability of wildlife monitoring efforts. We emphasized the importance of efficient hardware combined with edge monitoring in achieving the dream of a fully automated monitoring system in the near future.
Overall, we believe that the revised discussion section has enriched the theoretical interpretation of our study results and provided valuable insights into the impact factors, limitations, and prospects of using deep learning for intelligent automatic monitoring of waterbirds in surveillance videos.
Thank you again for your positive review and for taking the time to read and evaluate our manuscript.
Reviewer 3 Report
The manuscript discusses the importance of waterbird monitoring for wetland ecosystem conservation and management, particularly in China where improved wetland protection infrastructure has increased the need for data analysis techniques. The authors propose an improved detection method called YOLOv7-waterbird that uses deep learning techniques for real-time surveillance video analysis to identify attention regions and perform waterbird monitoring tasks. The method was tested on the Waterbird Dataset and achieved a mean average precision value of 67.3%, with better performance for small waterbird detection compared to the baseline model. The authors suggest that this algorithm could aid in wildlife conservation efforts by allowing for more accurate monitoring of waterbirds using existing surveillance cameras.
Despite the relatively low mean average precision achieved, this manuscript is paving the way for more extensive studies which will certainly reach a higher acurancy. The manuscript is well presented and the results are are robust. The conclusions are well supported by the results. I do not have suggestions for improving the manuscript as I judge it a nice piece of work.
Author Response
Response to reviewer3:
Comment 1:
The manuscript discusses the importance of waterbird monitoring for wetland ecosystem conservation and management, particularly in China where improved wetland protection infrastructure has increased the need for data analysis techniques. The authors propose an improved detection method called YOLOv7-waterbird that uses deep learning techniques for real-time surveillance video analysis to identify attention regions and perform waterbird monitoring tasks. The method was tested on the Waterbird Dataset and achieved a mean average precision value of 67.3%, with better performance for small waterbird detection compared to the baseline model. The authors suggest that this algorithm could aid in wildlife conservation efforts by allowing for more accurate monitoring of waterbirds using existing surveillance cameras.
Despite the relatively low mean average precision achieved, this manuscript is paving the way for more extensive studies which will certainly reach a higher acurancy. The manuscript is well presented and the results are are robust. The conclusions are well supported by the results. I do not have suggestions for improving the manuscript as I judge it a nice piece of work.
Response:
Thank you for your positive review and for taking the time to read and evaluate our manuscript, We will continue to work towards improving the accuracy of our method.